# Evidence-based intervention to reduce avoidable hospital admissions in care home residents (the Better Health in Residents in Care Homes (BHiRCH) study): protocol for a pilot cluster randomised trial

Elizabeth L Sampson,[1] Alexandra Feast,[1] Alan Blighe,[2] Katherine Froggatt,[3] Rachael Hunter,[4] Louise Marston,[4] Brendan McCormack,[5] Shirley Nurock,[1] Monica Panca,[4] Catherine Powell,[2] Greta Rait,[4] Louise Robinson,[6] Barbara Woodward-Carlton,[2] John Young,[7] Murna Downs[2]

For numbered affiliations see end of article.

**Correspondence to**
Dr Elizabeth L Sampson;
e.sampson@ucl.ac.uk

## ABSTRACT

**Introduction** Acute hospital admission is distressing for care home residents. Ambulatory care sensitive conditions, such as respiratory and urinary tract infections, are conditions that can cause unplanned hospital admission but may have been avoidable with timely detection and intervention in the community. The Better Health in Residents in Care Homes (BHiRCH) programme has feasibility tested and will pilot a multicomponent intervention to reduce these avoidable hospital admissions. The BHiRCH intervention comprises an early warning tool for noting changes in resident health, a care pathway (clinical guidance and decision support system) and a structured method for communicating with primary care, adapted for use in the care home. We use practice development champions to support implementation and embed changes in care.

**Methods and analysis** Cluster randomised pilot trial to test study procedures and indicate whether a further definitive trial is warranted. Fourteen care homes with nursing (nursing homes) will be randomly allocated to intervention (delivered at nursing home level) or control groups. Two nurses from each home become Practice Development Champions trained to implement the intervention, supported by a practice development support group. Data will be collected for 3 months preintervention, monthly during the 12-month intervention and 1 month after. Individual-level data includes resident, care partner and staff demographics, resident functional status, service use and quality of life (for health economic analysis) and the extent to which staff perceive the organisation supports person centred care. System-level data includes primary and secondary health services contacts (ie, general practitioner and hospital admissions). Process evaluation assesses intervention acceptability, feasibility, fidelity, ease of implementation in practice and study procedures (ie, consent and recruitment rates).

**Ethics and dissemination** Approved by Research Ethics Committee and the UK Health Research Authority. Findings

## Strengths and limitations of this study

► We are testing a complex intervention that has been developed to reduce avoidable acute hospital admissions in nursing home residents. The intervention has been developed in collaboration with key stakeholders, taking into account the complexities of implementing enhanced practice in this setting.

► A strength of the intervention is that implementation is led in the care home by Practice Development Champions, supported by practice development support group.

► We will assess feasibility of recruitment and outcome data collection at individual and system level and also consider how well the intervention was implemented in practice.

► This is a pilot study that will indicate whether a further study is warranted but will not give data on the effectiveness of the intervention.

will be disseminated via academic and policy conferences, peer-reviewed publications and social media (eg, Twitter).

**Trial registration number** ISRCTN74109734; Pre-results.

## INTRODUCTION
### Background
Currently in the UK 421 100 people aged over 65 years live in residential care, including care homes with nursing (referred to subsequently in this paper as 'nursing homes').[1] Older people living in nursing homes have increasing levels of comorbidity,[2–4] frailty and physical health needs.[5] Ambulatory care sensitive conditions (ACSCs) are 'conditions that can lead to unplanned hospital

admissions that may been have avoidable or manageable by timely access to medical care in the community'.[6] The conditions include: angina, asthma, cellulitis, chronic obstructive pulmonary disease, congestive heart failure, dehydration, diabetes mellitus, gastroenteritis, epilepsy, hypertension, hypoglycaemia, urinary tract infections (UTI), pneumonia, severe ear, nose and throat infections.[7]

In the UK, ACSCs account for one-sixth of hospital admissions from all age groups.[8] Our ageing population led to a 40% increase in admissions between 2001 and 2011,[9] and all-cause hospital admissions from nursing homes rose by 63% between 2011 and 2015.[10] Four ACSCs contribute to a large proportion of hospitalisations from nursing homes: respiratory infections,[11–14] acute exacerbation of chronic heart failure,[15 16] UTIs[13 17] and dehydration,[13] and may underlie other problems such as falls and delirium.

In addition to causing distress to residents, their families and staff, hospitalisation is expensive for health and social care systems. Hospital admission increases the risk of decline in functional ability, delirium, adverse events and prolonged stays.[18 19] Areas with many nursing homes tend to have higher rates of unplanned hospital admission in the over-75 age group.[20] The King's Fund[8] and British Geriatrics Society[2] have raised concerns about the quality of healthcare provision to nursing homes. The UK National Health Service (NHS)[21] has made reducing avoidable hospital admission from nursing homes a policy imperative.

A number of interventions have been developed, falling broadly into two categories; multicomponent interventions (implementation of a range of tools) and single-component interventions (predominantly advance care planning or single disease care pathways (eg, pneumonia). Multicomponent interventions show significant reductions in avoidable admissions.[22–26] Key characteristics include enhancing knowledge and skills of nursing home staff,[27] clinical guidance and decision-support tools (care pathways), engaging with families[28] and specialist input from geriatricians or nurse practitioners.[22] In addition, research highlights the importance of collaborative development of interventions with nursing home staff,[29] residents and families,[28] considering implementation support[16] and using local champions.

The intervention with the strongest evidence base is 'INTERACT' (Interventions to Reduce Acute Care Transfers). This complex intervention, developed and implemented in the USA, aims to detect and diagnose a range of medical conditions in residents recently discharged from hospital to skilled nursing facilities and reduce readmissions. It comprises a quality improvement programme focusing on the management of acute changes in residents' condition[16]:

► Communication tools, for example, Stop and Watch early warning tool and Situation Background Assessment Response (SBAR) structured communication with primary care tool.

► Care pathways or clinical tools addressing, for example, dehydration, UTI, fever and acute mental status change.

► Advanced care planning, tracking and communication tools.

### Intervention development
In the preliminary stages of the Better Health in Residents in Care Homes (BHiRCH) programme, we worked with stakeholders including staff and our carer reference panel (CRP), to develop and adapt the INTERACT tools for use in the UK (Blighe *et al*, unpublished data, 2018). We identified current UK care pathways for ACSCs in nursing homes for our four key conditions (respiratory infections, acute exacerbation of chronic heart failure, UTIs and dehydration) and adapted them in consensus co-design workshops with staff and care partners.

We conducted a rapid research review and semistructured interviews to understand the optimal approach to enhance skills and knowledge in nursing home staff. The Promoting Action on Research Implementation in Health Services (PARiHS) framework was used to co-design implementation support and guidance.[30] We conducted a feasibility and acceptability study in two nursing homes in the north of England between October 2016 and January 2017 to optimise the intervention and its implementation.

### Aim and objectives
The aim of this pilot trial is to indicate whether a definitive study is warranted.

### Primary objective
Indicate whether the intervention is acceptable and feasible.

### Secondary objectives
1. Establish whether consent procedures facilitate collection of sufficient individual-level data.
2. Assess the effectiveness of the implementation strategy.
3. Assess intervention fidelity.
4. Assess level of nursing home staff engagement with the intervention.
5. Explore whether the intervention would be sustainable outside the trial context.
6. Measure completeness of data collection, documentation, return rate of questionnaires and assess potential primary and secondary outcomes for a definitive trial.
7. Assess feasibility of collecting data for economic evaluation.

### METHODS AND ANALYSIS
### Trial design
A pilot cluster randomised trial in nursing homes with process evaluation. The intervention aims to deliver enhanced 'usual care', with specifically developed tools that formalise current good practice, delivered by

existing staff. The nursing home is the unit of allocation and intervention.

## Study population
### Nursing homes
We will recruit 14 nursing homes (eight in West Yorkshire and six in London) with adequate staffing to implement the intervention and support research activities. Nursing homes rated 'inadequate' by the Care Quality Commission (UK body responsible for assuring care quality) are ineligible. Nursing homes will be identified via local Clinical Research Networks and the Enabling Research in Care Homes Network. We will gain written permission from the manager, regional manager or owner. Our intervention will be implemented at nursing home level so individual consent is not required to receive the intervention.

### Individual participants
All English-speaking staff and residents over 65 years and their care partners (family members or friends) will be invited to participate in the collection of individual level data until we have recruited approximately 20 residents and staff from each nursing home. Residents receiving end-of-life care, who are under 65, non-English speaking or have stated they do not wish to be involved in research will be excluded. No further data will be collected from the care partner if the resident dies during follow-up.

## Sample size and selection
This is a pilot study to inform the sample size calculation for a definitive trial, so no sample size calculation was conducted. Nursing homes will be purposively selected including a range of providers (large and small chains, independent providers), urban, suburban and rural. Nursing homes will be randomised prior to intervention; four in West Yorkshire and three in Greater London (seven total) to the intervention and four in West Yorkshire and three in Greater London (seven total) to 'usual care' stratified by location, by Priment Clinical Trials Unit using a randomisation list drawn up by an independent statistician.

## Consent procedures
### Residents
The care home manager or deputy manager will identify all potentially eligible residents. If necessary, the research team will conduct a capacity assessment with respect to the participation in this trial, adhering to the Mental Capacity Act (2005).

If the resident lacks capacity to consent, we will use a personal consultee (friend or family), or if none is available, a professional consultee. This will be a member of health or social care staff with a professional relationship to the resident but no connection with the project. If a resident loses capacity during the study a consultee (either personal or professional) will be found. During the study, we will consider ongoing (process) consent,

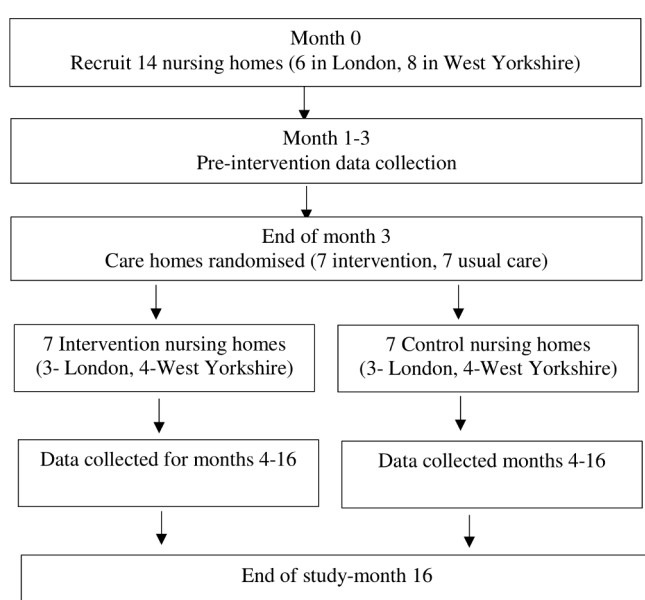

**Figure 1** Pilot trial flow chart.

checking willingness to participate with the resident or consultee.

### Care partners and nursing home staff
Care partners associated with residents who have been recruited to the study and nursing home staff will be asked if they wish to answer questionnaires and/or take part in qualitative interviews and give informed consent for this.

## Study procedures
The study will run for 16 months between 1 November 2017 and 3 March 2019 (see figure 1). Posters will be placed in nursing homes and there will be sign-up sheets where potential participants can indicate interest. We will publicise the project using established communications between the nursing home and care partners such as their regular newsletters. A 'launch' event will be organised in each nursing home where the research team, members of the CRPs and nursing home staff will explain the study and distribute recruitment literature.

Each participating nursing home will appoint a research facilitator to support the research team with recruitment activities, ensuring nursing home-level data collected without consent from the resident are pseudoanonymised prior to being given to the research team. The research facilitator may be a care home nurse or administrator and will be supported by the research assistant allocated to the care home. They will not be involved in implementing the intervention.

## INTERVENTION
This intervention is delivered by nursing home staff trained and supported by Practice Development Champions (PDCs) nominated from each intervention nursing home by its managers, based on a person specification

## Box 1 Practice Development Champion person specification

The Practice Development Champion will:
► Be a registered nurse.
► Have been working in the nursing home for at least 6 months.

When selecting a Practice Development Champion, we are looking for someone who:
► Has some knowledge of good practice in supporting healthcare and has an interest in the topic *(can demonstrate some essential knowledge of the management of the four conditions: chronic heart failure, respiratory infections, urinary tract infection, dehydration)*.
► Knows co-workers *(has been in the organisation long enough to know the staff and how they work)*.
► Knows the environment *(has some insight into the culture of the setting)*.
► Knows the organisation *(knows their way around the organisation, for example, who's who, policies in place, decision-making structures)*.
► Possesses effective communication skills *(could include attributes of being open minded, being creative, has experience of managing meetings/groups, able to talk in front of groups)*.
► Is self-aware and resilient *(has insight into their support needs, but is also not afraid of challenge/conflict; willing to engage in own professional development)*.
► Is reliable and dependable *(has time they can dedicate to this work (in writing from their manager); carries through with responsibilities, meets deadlines or negotiates otherwise; is not intending to be on extended leave during intervention period)*.
► Is respected by co-workers *(has a good relationship with co-workers which means they will be listened to with respect to new ideas)*.

These criteria are ESSENTIAL and are NOT listed in a hierarchy/order of importance, that is, they are all equally important.

From Seers *et al* 2012 FIRE (facilitating implementation of research evidence): a study protocol http://implementationscience.biomedcentral.com/articles/10.1186/1748-5908-7-25

developed from research on facilitating implementation of research evidence (box 1).

### Implementation support
#### PDCs workshop

A 1-day workshop will be delivered for the two PDCs from each nursing home by the research team comprising an introduction to the four key conditions (respiratory infections and UTIs, dehydration and acute exacerbation of chronic heart failure) and elements of how to bring about organisational change. This will focus on: how to establish and coordinate the practice development support group of nursing home staff, care partners and external staff (eg, primary care professionals) who can support introduction and embedding of the change, strategies for engaging people and encouraging continued participation and strategies for gathering routinely collected data to monitor implementation. They will be given an overview of the intervention materials including the structured approach to effective communication with primary care staff. We will also explore potential changes to communication flows and recording of information about residents.

#### Ongoing implementation support

1. PDCs will be supported by a project handbook created for staff use. In addition, the *Practice Development Workbook for Nursing, Health and Social Care Teams: Resources for Health and Social Care Teams* (Dewing *et al*, 2014) will be provided. PDCs can access peer support on the study website and monthly telephone support from the research team.
2. PDCs will select members of a practice development support group to support their work in the nursing home. This Quality Collaborative approach[31] involves diverse stakeholders working together to close the gap between actual and potential practice. The group will support the PDCs and intervention implementation taking into account the context at their site.
3. Monthly support phone call between PDCs and a senior nurse researcher.

### The intervention

This commences after recruitment and randomisation. There are three key components, all paper based as UK care homes have variable use of electronic records:

1. Early Warning Tool (*Stop and Watch Early Warning Tool;* http://www.pathway-interact.com/wp-content/uploads/2017/04/148604-Stop-and-Watch-v4_0.pdf).
2. *Care Pathway* (clinical guidance and decision support system).
3. Structured method for communicating with primary care (*SBAR*).

#### Stop and Watch Early Warning Tool

Care assistants or nurses will use this when they or anyone else in the nursing home (including other staff and care partners) notices a change in a resident, at latest by the end of the shift. They will circle observed changes, notifying the nurse and giving them the completed tool to be placed in the nursing home records.

#### Care Pathway

This is a two-step clinical guidance and decision support system. The initial 'Primary' assessment comprises screening questions with the potential to trigger a more detailed 'Secondary' assessment. If the Primary or Secondary Assessment result is ambiguous, the *Care Pathway* will be administered at 6 hour intervals, until concerns have resolved and/or appropriate intervention has been instigated. The nurse will record the outcome of the primary and secondary assessment and their care plan in the residents' care records and make a clinical decision about the next course of action which may include direct further monitoring using the *Stop and Watch Early Warning Tool*, initiating treatment in the nursing home, or if the assessment indicates a potential diagnosis or immediate concern, communication with primary care using the *SBAR* process. The nurse will feed back on the course of action to relevant staff on each shift, domestic staff and care partners, as appropriate. Copies of the completed *Care Pathway* will be kept with the resident's record.

## The SBAR method

A structured method for communicating critical information to primary care. The nurse uses this when they want primary care input for one of their residents after the *Care Pathway* indicates a risk of decline. Before calling, the nurse will organise the briefing information on paper using the four elements (situation, background, assessment and recommendation).

## Usual care group

Participants in nursing homes randomised to control arms will receive usual care according to existing local policy and practice. All medications and treatments will be permitted.

## Study measures

We will collect data in three domains: (1) individual level data on nursing home residents, their care partners and staff; (2) system-level data and (3) process data.

### Individual-level data

These will be collected from staff, care partners or residents who have given informed consent or from residents for whom we have obtained agreement from a consultee. These data includes resident, care partner and staff demographics, assessment of resident functional status (Barthel index[32]) and resident service use and quality of life to inform health economic analysis (Client Services Receipt Inventory[33] (CSRI), EQ-5D-5L[34]). Staff education, role and the extent to which they perceive the organisation to support person-centred care (Organisational Support for Person-centred Care Assessment Tool[35]) and Nurse Ratings of Communication with Primary Care Questionnaire[36] will be gathered to understand the context (table 1).

### System-level data

These will be collected from the nursing home using existing paper or electronic record systems. Data will be pseudoanonymised and provided by the nursing home manager or research facilitator. We will document the total number of contacts with GPs, ambulances, A&E visits, deaths, staff turnover and the number of nursing home beds available to new residents in the previous month, per care home. We will document all hospital admissions and select a sample of 30 of these (where we have individual consent from a resident to access their health records), purposively selected to represent a range of underlying ACSCs diagnoses. We will explore the 'avoidability' of these with the Structured Implicit Record Review tool.[37] This facilitates comparison by two independent clinical experts (geriatrician and community nurse) of selected content in residents' nursing home records to assess whether the admission was 'avoidable'.

## Process evaluation

We will examine how the intervention is implemented in practice, collecting information on the number of 'Stop and Watch' Early Warning Tools completed, who noticed the change in the resident's health status, who completed the form, and actions that occurred after the form was completed; number of primary and secondary assessments, and outcome of assessments including the number of residents who underwent further monitoring, had their treatment initiated in the nursing home, or were referred to primary care. We will monitor intervention fidelity noting where nursing homes make amendments to the structure or content of the care pathway. PDCs will have a monthly support phone call with the research team to reflect on their activities and achievements which we will document. They will keep an activity log of support provided by practice development support groups to document the level of facilitation required to support the implementation process. To understand the effectiveness of study procedures we will collect data on consent and recruitment rates, the numbers of care partners who wish to be involved in the resident's care, assess completeness of outcome measures, data collection and return rate of questionnaires.

We will conduct 20 semistructured interviews (30–45 min each) with five nursing home managers, five nurses, five care assistants and five care partners from the seven intervention homes. We will explore participants' views on the effectiveness of the intervention in preventing avoidable hospital admissions and their experiences of implementing the intervention. Care partners will be purposively sampled to ensure a range of gender, age and types of family carer. All participants will give informed consent, including for recording of their interviews.

## Data management

All data will be entered onto paper case report forms (CRFs) and then into an encrypted password protected database in accordance with the UK Data Protection Act and General Data Protection Regulation. CRFs will not bear the participant's name but a pseudoanonymised identification number. To maintain confidentiality the research facilitator will remove the name of the participant from completed Stop and Watch Early Warning Tool and Care Pathway Primary and Secondary Assessment Forms and replace this with an ID number. Prior to analysis we will follow a standardised process for database lock.

## Randomisation

The SAS statistical programme (version 9.4) will be used for randomisation. Blinding is not feasible for research staff collecting data, but statisticians and health economists will be blinded to allocation. The randomisation variable will be supplied to them unlabelled, and main analysis completed using this.

## Data analysis plan

We shall follow Consolidated Standards of Reporting Trials (CONSORT) guidelines for reporting randomised trials. Given this is a pilot study, analyses will be mainly descriptive focussing on recruitment, participant characteristics,

**Table 1** Summary of data collected, outcome measures and time schedule

| | Data collected and tool used | Pre-intervention | Monthly | At 6 months only | Post-intervention |
|---|---|---|---|---|---|
| **Resident** | | | | | |
| Sociodemographics | Age, gender, ethnicity, marital status, highest level of education. | R | – | – | – |
| Service use in the prior month | Client Service Receipt Inventory (ref). Calculates service and total care costs. | R | R | R | R |
| Functional status | The Barthel Index.[32] | R | – | R | R |
| Resident quality of life-self rated | EQ-5D-5L (ref) self-rated health index and Visual Analogue Scale of current health state. | P | – | P | P |
| Resident quality of life-proxy rated | EQ-5d-Proxy (ref) care partner or staff member view of the resident's quality of life. | CP/S | – | CP/S | CP/S |
| **Care partner** | | | | | |
| Sociodemographics | Age, gender, ethnicity, marital status, years of schooling, highest level of education. | CP | – | CP | CP |
| Quality of life | EQ-5D-5L EuroQol (1990). | CP | – | CP | CP |
| Preferred role | How much and how they like to be involved in the residents care. | CP | – | – | – |
| **Staff** | | | | | |
| Staff sociodemographics | Age, gender, ethnicity, number of years of education. | R | – | – | – |
| Staff work characteristics | Highest qualification, role in care home, length of service, shift pattern, first language. | R | – | – | – |
| Organisational support for person-centred care | The Person-Centred Care Assessment Tool (ref). | S | – | S | S |
| Communication with primary care | Nurse-General Practitioner Communication Needs Assessment Questionnaire. | S | – | S | S |
| Perceived knowledge and skills for early detection in changes in health | Developed from feasibility study. Assesses key knowledge and skills needed to implement the intervention. Rated on 5-point Likert scale; 1 (disagree completely) to 5 (agree completely). | S | – | S | S |
| **System-level data** | | | | | |
| Number of hospital admissions | Respiratory infections, urinary tract infections, dehydration, congestive heart failure? | S | S | S | S |
| 'Avoidability' of admissions | Structured Implicit Record Review. Saliba *et al*, 2000 | S | S | S | S |
| Use of primary assessment tool | Respiratory infections, urinary tract infections, dehydration, congestive heart failure? | S | S | S | S |
| Use of secondary assessment tool | Respiratory infections, urinary tract infections, dehydration, congestive heart failure? | S | S | S | S |
| Out of hours GP contacts | GP visits or telephone contact. | S | S | S | S |
| Ambulances and hospital use | Number and length of hospital admissions (days), accident and emergency attendances and readmissions. | S | S | S | S |

Continued

**Table 1** Continued

| | Data collected and tool used | Pre-intervention | Monthly | At 6 months only | Post-intervention |
|---|---|---|---|---|---|
| Deaths in the last calendar month | | S | S | S | S |
| Staff turnover | | S | S | S | S |
| Care home occupancy level | Number of available beds to new residents. | S | S | S | S |

CP, care partner; P, participant; R researcher; S, care home staff.

other baseline and outcome variables, loss to follow-up and tabulation of serious adverse events (SAEs). We shall compare rates of hospital admission for ACSCs and other outcomes between the control and intervention groups and calculate 95% CIs. These, along with estimates of SD of other outcome measures and intraclass correlation coefficients will inform the sample size calculation for a full trial. We will summarise completeness of data collection on outcome measures and, for questionnaires, describe distributions and response rates.

### Economic evaluation

Analyses will conform to accepted economic evaluation methods (National Institute for Health and Care Excellence, 2008). We will calculate costs associated with the intervention, including the cost of enhancing staff's knowledge and skills and resources associated with implementation. Resource use associated with hospital admissions, primary care and other NHS and social care costs will be collected using the CSRI. Costs will be reported from an NHS/prescribed specialised services (PSS), government and societal perspective. We will assess feasibility of calculating quality-adjusted life years (QALYs) for residents, using the EQ-5D-5L. In the societal level analysis, a calculation of carers' QALYs will be included. We will provide an initial estimate of the incremental mean cost per QALY gained in intervention compared with control homes. Cost-effectiveness acceptability curves, showing the percentage of cases that the intervention is cost-effective over a range of values of willingness to pay for a QALY gained, will be constructed for each different costing perspective and for the different methods of calculating QALYs. We will model the lifetime costs and outcomes of the intervention compared with controls. This will involve assessing the quality of the published information available, the development of an initial model and identification of which cost and outcome components would benefit most from further research (ie, extra value of perfect information and extra value of partial perfect information analysis).[38 39]

### Qualitative methods

A verbatim transcript of qualitative interviews will be made and entered into qualitative analysis software (NVIVO) and key themes coded using framework analysis.[40] A sample of interviews will be analysed by two senior investigators to check levels of coding agreement with the template.

### Public and patient involvement

The original research proposal was developed in collaboration with UK Dementia and Neurodegenerative Diseases Network Patient and Public Involvement representatives (SN and BW-C), who are grant co-applicants. Two CRPs have been created to ensure public involvement at all stages, chaired by SN (London) and BW-C (Bradford). Each panel comprises up to eight family carers of people with dementia and a person living with dementia. The carers are members of, and supported by, the Alzheimer's Society research volunteer network. They will meet at 6-monthly intervals throughout the programme, advise and work collaboratively on recruitment and consent processes, accessibility of information leaflets, data collection, interpretation and dissemination. Our CRP chairs will attend study recruitment and information events at participating care homes to publicise the trial. They provide strategic oversight by attending programme steering group and international advisory meetings. We held informal consultations with a group of seven residents living in one nursing home to inform development of study information materials for residents.

### Monitoring and trial management

Our intervention is low-risk and an enhancement of usual care. We will collect data on potential SAEs; those that may result in death, be life-threatening, require hospitalisation or prolongation of existing hospitalisation or result in persistent or significant disability or incapacity. These will be reported to the chief investigator and escalated to the research ethics committee if they are unexpected and related to research procedures.

A trial management group meets monthly and includes individuals responsible for day-to-day study management. It will monitor all aspects of the conduct and progress of the study, ensuring adherence to the protocol and will take appropriate action to safeguard participants and trial quality. For this pilot trial, it also takes on the role of a data monitoring committee. The BHiRCH Programme Steering Committee includes senior members of the research team and external experts and supervises the overall programme, on behalf of NIHR and the Sponsor. An international advisory group provides advice and

guidance to ensure the project remains grounded in real experience and is informed by international best practice and research.

## DISCUSSION

The aim of our study is to pilot test an intervention to reduce avoidable acute hospital admissions for the four most common ACSCs in UK nursing home residents. Our intervention is closely aligned with UK health policy which aims to improve healthcare provided within nursing homes and prevent distressing transitions to acute hospitals. Despite policy imperatives, little empirical research has been conducted in this area in the UK. The quality of intervention studies has varied with insufficient attention paid to methodology, particularly implementation, intervention adherence and clustering effects.

We have adapted elements of an existing intervention (INTERACT) to the UK context with stakeholder input and extensive co-design. Trials of complex interventions are often methodologically challenging but the nursing home environment brings additional issues. Each nursing home, even those that are part of large company chains, is unique in context, with a distinct culture of care and variability in practice which may be greater than that seen in clinical NHS settings. A particular strength of our intervention is that the implementation is informed by implementation science theory, tailored to the context and led by local PDCs supported by a local PDC support group comprised of key stakeholders who can assist with achieving change in care practice.

We anticipate participant recruitment, particularly of residents with impaired capacity to consent to research procedures may be challenging. Additionally, nursing homes may not have the infra-structure to support data collection . We plan to overcome these problems through developing the role of an existing member of the nursing home staff as a research facilitator and will provide reimbursement for their time and additional work.

Our pilot study will provide data to indicate whether a fully powered randomised controlled trial is warranted. In addition it will provide invaluable information on how to optimise the implementation of complex interventions in nursing homes.

## ETHICS AND DISSEMINATION

Some residents may have dementia or other conditions that impair their ability to give informed consent to participate in the individual data collection, and we will adhere to the UK Mental Capacity Act (2005). We have an established procedure should we observe incidents of concern and will adhere to local authority safeguarding procedures.

Academic dissemination will be through publication of results, reported per CONSORT guidelines in international peer-review journals and conference presentations. Regular updates will be given via social media, our website

(https://www.brad.ac.uk/health/dementia/research/bhirch/) and twitter (@BHIRCHCareHomes). We will present at congresses attended by nursing home owners and staff and will hold a national conference at the end of the study to disseminate findings and share best practice on active care for ACSCs.

**Author affiliations**
[1]Marie Curie Palliative Care Research Department, Division of Psychiatry, University College London, London, UK
[2]Centre for Applied Dementia Studies, University of Bradford, Bradford, West Yorkshire, UK
[3]International Observatory on End of Life Care, Lancaster University, Lancaster, UK
[4]Research Dept of Primary Care and Population Health and PRIMENT Clinical Trials Unit, University College London, London, UK
[5]School of Health Studies, Queen Margaret University, Edinburgh, UK
[6]Institute for Health and Society and Newcastle University Institute for Ageing, Newcastle University, Newcastle, UK
[7]Academic Unit of Elderly Care and Rehabilitation, Bradford Institute for Health Research, University of Leeds, Bradford, UK

**Acknowledgements** The authors would like to thank the members of our carer reference panels and the Alzheimer's Society research network volunteers. We are grateful to the nursing homes that supported our feasibility study.

**Contributors** ELS, AF, AB, KF, RH, LM, BMcC, SN, MP, CP, GR, LR, BW-C, JY, MD were involved in development of the intervention and made substantial contributions to the concept and design of the study and protocol. ELS and AF wrote the protocol paper. ELS, AF, AB, KF, RH, LM, BMcC, SN, MP, CP, GR, LR, BW-C, JY, MD critically revised the manuscript and approved the final version.

**Funding** This work was supported by the UK NIHR grant number RP-PG-0612-20010.

**Disclaimer** The views expressed are those of the author(s) and not necessarily those of the NHS, the NIHR or the Department of Health and Social Care.

**Competing interests** None declared.

**Ethics approval** London Queen Square Research Ethics Committee (Reference: 17/LO/1542) and the UK Health Research Authority.

**Provenance and peer review** Not commissioned; externally peer reviewed.

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
