## [Reviewer comments · BMJ Open]

ARTICLE DETAILS

TITLE (PROVISIONAL)	An evidence-based intervention to reduce avoidable hospital admissions in care home residents (the Better Health in Residents in Care Homes- BHiRCH study); protocol for a pilot cluster randomised trial
AUTHORS	Sampson, Elizabeth; Feast, Alexandra; Blighe, Alan; Froggatt, Katherine; Hunter, Rachael; Marston, Louise; McCormack, Brendan; Nurock, Shirley; Panca, Monica; Powell, Catherine; Rait, Greta; Robinson, Louise; Woodward-Carlton, Barbara; Young, John; Downs, Murna

VERSION 1 - REVIEW

REVIEWER	Sarah Purdy University of Bristol UK
REVIEW RETURNED	27-Sep-2018

GENERAL COMMENTS	Thank you for asking me to review this well constructed and clearly written paper. It outlines a protocol for a feasibility study for an RCT of a complex intervention to reduce acute admissions from nursing care homes. The paper is easy to follow and outlines the key evidence and development and components of the intervention. As the protocol is for a pilot/feasibility study there is minimal underpinning data to support the paper (unlike a full trial protocol) this makes it less informative for the reader. In terms of the future study (and a full trial) the importance of delivery of and fidelity with the intervention cannot be overemphasized - these sorts of studies often fall down on this aspect of the research.
---

REVIEWER	Greta Cummings & Kaitlyn Tate (doctoral student) University of Alberta, Canada
REVIEW RETURNED	05-Oct-2018

GENERAL COMMENTS	This is a protocol paper for a pilot of a multi-component intervention to reduce avoidable hospital admissions for care home residents (the Better Health in Residents in Care Homes (BHiRCH) programme). Generally, more clarification is needed on the procedures in the actual protocol – it is unclear how the intervention will be implemented and evaluated.
--

	1. Introduction a. Please clarify if “one sixth of hospital admissions” (p 3, line 22) refers to older populations in the UK or all populations. b. Is the assumption that avoidable hospitalizations are those caused by or related to Ambulatory Care Sensitive Conditions? If so, this needs to be explicit and clearly rationalized. It’s unclear if these are just examples. How were the four key conditions identified and how are you determining if transfers are clearly related to only these conditions (preferably prospectively)? c. Objective 7 should read “Assess” (p 5, line 6). 2. Sample Size and Selection. More information needed around the purposive selection and randomization process. Will you match residents based on characteristics, acuity for example? 3. Consent procedures (p 6, line 6). Which member of the nursing home staff will identify potentially eligible residents (i.e will a health care aide, registered or licensed practical nurse, manager?) Will it be the same person for all residents? 4. Care partners and nursing home staff (recruitment; p 6, line 24-27). Will you only approach care partners after resident agrees to participate in the study? 5. Study procedures. a. Recruitment procedures are very unclear. How will you avoid bias in selection? Will you approach all residents who fit criteria for consent before the transition is needed? How will participants be contacted after they sign up? How will their contact information be kept confidential? b. Is 16 months for the intervention portion only, or the whole study? (p 6, line 31). c. The term “family members” is used on p 6, line 34 – is this the same as care partners? d. Please define/describe Carer Reference Panels (p 6, line 37). e. Please describe the role and training of the Research Facilitator (will it be a front-line staff member, manager?) 6. Intervention. a. When referring to “each nursing home” (p 6, line 52), do you mean in the intervention and control group or only in the intervention group? 7. Implementation support. a. There are a lot of topics to cover in the one day workshop. Who is all expected to attend this workshop? Will they be evaluated on their knowledge after the workshop? b. How will implementation be supported regarding issues around intervention fidelity, attrition of participants etc.? 8. The intervention (p 7-8). a. I would expect to see the proposed analysis of the intervention implementation and outcomes. As this is a large, complex intervention, it is important to describe how fidelity across sites will be assessed, how they will deal with staff attrition and education. b. Will health care aides be trained to use the Stop and Watch Early Warning Tool? If the tool is to equip HCAs with a way to communicate ambiguous or early changes in condition, then why
--	---

	are nurses filling it out in your study? Will the tool be paper or electronic? c. Will all nurses be trained on recording the outcome of Primary and Secondary Assessments/care plans and other steps? Or just RNs, just LPNs? 9. Individual level data. What if staff refuse to consent or participate in their study home? How will inconsistencies between who is participating for different transfers be addressed? 10. System level data (p 9). a. Will the research facilitator collect the data from existing records? Or a research assistant? Will this person be trained? Will this data be from paper charts or electronic documentation? (line 4) b. Will the number of contacts tracked be per resident or per facility? (line 7) c. How are you determining “clinical experts”? Will these experts be from the research team or included facilities? 11. Process evaluation. Will the research team be able to determine if the Stop and Watch Tool should have been completed, but wasn't (i.e. if the staff missed an important condition change)? Again, how will fidelity issues will be addressed needs to be described – what elements need to be consistent, why and how will you ensure that or consider that in your evaluation? 12. Data management. Why is data being entered into paper case reports? Why not a secure, electronic report? (p 9, line 53). 13. Monitoring and trial management. Will data collected be analyzed and reported? A detailed analytic plan needs to include what relationships they are going to text for and these should also all align with hypotheses, which are note stated. 14. What will procedures be adapted from or based on (experiences, empirical data)? There is no explanation of the proposed analysis, despite the robust information that is going to be collected from staff, residents and the organization.
--	--

REVIEWER	A/Prof Magnolia Cardona Centre for Research in Evidence Based Practice; Faculty of Health Sciences and Medicine; Bond University, QLD, Australia
REVIEW RETURNED	07-Oct-2018

GENERAL COMMENTS	Thank you for the opportunity to review your work addressing a very important need. The manuscript is well written and the scope of the multicomponent intervention looks comprehensive; activities are supported by evidence, the study will use previously tested instruments for delivering the intervention and measuring outcomes, and follow CONSORT guidelines for reporting; and the evaluation activities cater for system and user perspectives. The protocol has sufficient level of detail to allow replication by others. The researchers' efforts to contribute empirical evidence in this area where previous implementation methods have not been satisfactory is commended.
--

However, several issues raise questions. For instance, it is unclear how the sample of 14 nursing homes for this pilot was selected to obtain the representativeness of size, location and provider types. For practical purposes, this is important to know even if this is a pilot.

I found it confusing that the intervention would be conducted without individual resident consent since it has the potential to cause distress to individuals potentially wanting a hospital transfer. The fact that a cluster RCT has the care home as the unit of analysis does not mean individual consent can be waived. Yet, it appears that care partners and nursing home staff's consent will be obtained to participate in interviews and assessing quality of life of the resident. This inconsistency needs to be clarified, even if the ethics approval has already been granted under those conditions. This intervention in humans would not pass ethics approval without individual resident consent elsewhere since it goes beyond the quality improvement purpose; the care pathway decisions have implications for the individual.

The need for a 12-month intervention is questionable if this is a pilot, even if staff are allowed to stop the activity if they feel uncomfortable (page 49). Testing of procedures for feasibility and fidelity could be completed within 6 months with the pre and post intervention data collection added at each end of the intervention. By the way, the wording in the last box of Figure 1 (page 21) is confusing. The duration of process should be consistent with the more clear explanation on page 41.

A major concern is the exhaustive patient-level and system-level data documentation required to implement and evaluate the program. On page 48 the authors argue that the intervention is an enhancement of usual clinical care, and that (page 49) resident assessment has been kept as brief as possible to minimise potential burden. While one of the secondary objectives is to explore the sustainability outside trial context, the translatability is questionable at the outset as the time demands of these steps are unrealistic in the course of routine care: identifying deterioration, repeating assessments at 6-hourly intervals, communicating actions with primary care (18-item questionnaire) and documenting detailed outcomes including 'avoidability of hospital admissions' engaging expert clinical adjudication panels, and procedures for reporting serious adverse events. The exhaustive set is unlikely to be embedded in routine practice in a non-hospital and non-research environment; this extends to whether equivalent PDC personnel can be found across facilities in a real-life larger study. Knowledge translation principles require affordability, feasibility and consideration of risks, not just evidence of technical effectiveness. It's acknowledged that this study aims to find out "how to optimise the implementation of complex interventions in nursing homes" but I would warn that the excessive demands on staff carry the risk of turning the multicomponent intervention into an academic exercise with limited applicability in a chaotic and under-resourced real-life world. The discussion should include this as a limitation of the proposed study.

VERSION 1 – AUTHOR RESPONSE

Reviewer comment	Response
Reviewer 1	
As the protocol is for a pilot/feasibility study there is minimal underpinning data to support the paper (unlike a full trial protocol) this makes it less informative for the reader. In terms of the future study (and a full trial) the importance of delivery of and fidelity with the intervention cannot be overemphasized	Thank you for your comments. We acknowledge this is a protocol for a pilot study but believe it is still important to publish developmental work. We agree, hence the conduct of this pilot work
Reviewer 2	
Generally, more clarification is needed on the procedures in the actual protocol – it is unclear how the intervention will be implemented and evaluated.	Thank you for the careful reading of our paper. We hope we have clarified the procedures within the limits of the journal word count) by answering the queries below and making the necessary amendments.
1. Introduction a. Please clarify if “one sixth of hospital admissions” (p 3, line 22) refers to older populations in the UK or all populations. b. Is the assumption that avoidable hospitalizations are those caused by or related to Ambulatory Care Sensitive Conditions? If so, this needs to be explicit and clearly rationalized. It’s unclear if these are just examples. How were the four key conditions identified and how are you determining if transfers are clearly related to only these conditions (preferably prospectively)? c. Objective 7 should read “Assess” (p 5, line 6).	This is across all age groups- amendment made (page 3 para 2). The definition of ACSC is given clearly on page 3 para 1. ACSCs are “are conditions that can lead to unplanned hospital admissions that may be have avoidable or manageable by timely access to medical care in the community”. This is an internationally operationalised definition as referenced in our paper by Grabowski (reference 7). Our four key conditions were identified from review of the literature- see page 3, paragraph 2, references 11-17. This has been corrected
2. Sample Size and Selection. More information needed around the purposive selection and randomization process. Will you match residents based on characteristics, acuity for example?	This is a pilot cluster trial and so we did not “match residents”. The randomization process is described in detail on page 5 para 5 and page 10 para 2.
3. Consent procedures (p 6, line 6). Which member of the nursing home staff will identify potentially eligible residents. Will it be the same person for all residents?	We have been more specific “The care home manager or deputy manager” page 6, para 1
4. Recruitment; p 6, line 24-27). Will you only approach care partners after resident agrees to participate in the study?	Care partners are recruited after the resident has been recruited. Some residents may have the capacity to agree to participant, others may not have the capacity to give informed consent to participate in research and the care partner may act as proxy and give their agreement (UK term for proxy consent). We have amended page 6 para 3 to make this clearer
5. Study procedures.	

a. Recruitment procedures are very unclear. How will you avoid bias in selection? Will you approach all residents who fit criteria for consent before the transition is needed? How will participants be contacted after they sign up? How will their contact information be kept confidential? b. Is 16 months for the intervention portion only, or the whole study? (p 6, line 31). c. The term “family members” is used on p 6, line 34 – is this the same as care partners? d. Please define/describe Carer Reference Panels (p 6, line 37). e. Please describe the role and training of the Research Facilitator (will it be a front-line staff member, manager?)	a. We will try and avoid bias by approaching all potentially eligible participants (page 6 para 1). All information is kept confidential as described in the paragraph on UK data protection rules and in detail on page 9 para 4. b. This is for the whole added study and we have clarified by reference to figure 1 c. Corrected to “care partners” d. These are described in detail on page 11 para 2 e. More detail has been added page 6 para 5
6. Intervention. a. When referring to “each nursing home” (p 6, line 52), do you mean in the intervention and control group or only in the intervention group?	a. We have clarified that these are the intervention homes (page 6 para 6)
7. Implementation support. a. There are a lot of topics to cover in the one day workshop. Who is expected to attend this workshop? Will they be evaluated on their knowledge after the workshop? b. How will implementation be supported regarding issues around intervention fidelity, attrition of participants etc.?	a. As stated on page 7 para 1, these workshops are attended by Practice Development Champions. The workshop is not ended with a “test” b. Ongoing implementation support is offered via a monthly phone call with an experienced nurse researcher- this information has been added (page 7 para 2). Monitoring of attrition is not part of intervention implementation. Attrition is an outcome measure as part of the pilot study process, which has an aim to establish whether sufficient individual- level data can be collected.
8. The intervention (p 7-8). a. I would expect to see the proposed analysis of the intervention implementation and outcomes. As this is a large, complex intervention, it is important to describe how fidelity across sites will be assessed, how they will deal with staff attrition and education. b. Will health care aides be trained to use the Stop and Watch Early Warning Tool? If the tool is to equip HCAs with a way to communicate ambiguous or early changes in condition, then why are nurses filling it out in your study? Will the tool be paper or electronic? c. Will all nurses be trained on recording the outcome of Primary and Secondary Assessments/care plans and other steps? Or just RNs, just LPNs?	a. This is a pilot study so we aim to measure whether sufficient individual outcome data can be collected. Thus, in the context of a pilot study, attrition rates are findings rather than something to be managed. Fidelity is assessed as described in the paragraph on process evaluation (page 9 para 2) b. in the UK context “health care aides” are care assistants and as described on page 7 para 4, both nurses or care assistant will use the tool. The tool is paper-based and we have added a rationale for this (page 7, para 3) c. The nurses records the primary or secondary assessment as stated on page 8 para 1.
9. Individual level data. What if staff refuse to consent or participate in their study home? How will inconsistencies between who is participating for different transfers be addressed?	This is a pilot study and so if a nurse refuses to consent or participate, this is valuable outcome data (see study aims, secondary objectives page 4, para 5)

10. System level data (p 9). a. Will the research facilitator collect the data from existing records? Or a research assistant? Will this person be trained? Will this data be from paper charts or electronic documentation? (line 4) b. Will the number of contacts tracked be per resident or per facility? (line 7) c. How are you determining “clinical experts”? Will these experts be from the research team or included facilities?	a. Data will be collected via existing paper or electronic records, depending on what are used in the care home (clarified on page 9 para 1) b. It is per care home- we have clarified this page 9 para 1 c. These are independent local senior clinicians including geriatrician and senior nurse. We have added more detail on this (page 9 para 2)
11. Process evaluation. Will the research team be able to determine if the Stop and Watch Tool should have been completed, but wasn't (i.e. if the staff missed an important condition change)? Again, how will fidelity issues will be addressed needs to be described – what elements need to be consistent, why and how will you ensure that or consider that in your evaluation?	We cannot determine if the Stop and watch tool should have been completed but wasn't. We do not have the clinical research capacity to independently assess each individual care home resident for changes in health condition on a daily basis. Process evaluation and fidelity are described in detail on page 9 para 2. This is a pilot study so we will be exploring whether it's possible to examine fidelity rather than absolute fidelity to the intervention.
Data management. Why is data being entered into paper case reports? Why not a secure, electronic report? (p 9, line 53).	When this study was set up, UK ethics committee preferred the use of a paper source document (case report form)
13. Monitoring and trial management. A detailed analytic plan needs to include what relationships they are going to test for and these should also all align with hypotheses, which are not stated.	This is a pilot study and is not powered to test study hypotheses or relationships. The analysis plan was written by an experienced statistician from a registered clinical trials unit as per standard guidance for pilot studies.
14. What will procedures be adapted from or based on (experiences, empirical data)? There is no explanation of the proposed analysis, despite the robust information that is going to be collected from staff, residents and the organization.	Please see comment above. The analysis plan is given on page 10 para 3.
Reviewer 3	
Thank you for the opportunity to review your work addressing a very important need. The manuscript is well written The researchers' efforts to contribute empirical evidence in this area where previous implementation methods have not been satisfactory is commended.	Thank you for this comment
It is unclear how the sample of 14 nursing homes for this pilot was selected to obtain the representativeness of size, location and provider types.	This is a pilot study so we will attempt to recruit homes which fulfil a range of these criteria as stated . However, given this is a pilot study we are testing whether it's possible to recruit a range of homes, rather than trying to recruit a representative sample as would be desirable in a definitive RCT.
I found it confusing that the intervention would be conducted without individual resident consent since it has the potential to cause distress to individuals potentially wanting a hospital transfer. The fact that a cluster RCT has the care home as the unit of analysis does not mean individual consent can be	This study has been extensively reviewed by UK ethics committees under the governance of the UK Health Research Authority (UK-HRA). They agreed that cluster consent was appropriate for this type of intervention and other cluster methodologies.

waived. Yet, It appears that care partners and nursing home staff's consent will be obtained to participate in interviews and assessing quality of life of the resident. This inconsistency needs to be clarified, even if the ethics approval has already been granted under those conditions. This intervention in humans would not pass ethics approval without individual resident consent elsewhere since it goes beyond the quality improvement purpose; the care pathway decisions have implications for the individual.	Obviously, this may differ from other countries. Numerous similar studies are approved using this paradigm. We would be happy to provide the reviewer with a comprehensive list of exemplar cluster studies conducted in the UK if this provides more reassurance.
The need for a 12-month intervention is questionable if this is a pilot, even if staff are allowed to stop the activity if they feel uncomfortable (page 49). Testing of procedures for feasibility and fidelity could be completed within 6 months with the pre and post intervention data collection added at each end of the intervention.	In the UK there is great seasonal variability in admission rates from care homes to acute hospitals. The 12-month follow up was to allow us to collect data on admission events that reflected this and was approved by the UK National Institute for Health Research (NIHR) who funded the study.
Wording in the last box of Figure 1 (page 21) is confusing. The duration of process should be consistent with the more clear explanation on page 41.	This has been amended (figure 1)
A major concern is the exhaustive patient-level and system-level data documentation required to implement and evaluate the program. On page 48 the authors argue that the intervention is an enhancement of usual clinical care, and that (page 49) resident assessment has been kept as brief as possible to minimise potential burden. While one of the secondary objectives is to explore the sustainability outside trial context, the translatability is questionable at the outset as the time demands of these steps are unrealistic in the course of routine care: identifying deterioration, repeating assessments at 6-hourly intervals, communicating actions with primary care (18-item questionnaire) and documenting detailed outcomes including 'avoidability of hospital admissions' engaging expert clinical adjudication panels, and procedures for reporting serious adverse events. The exhaustive set is unlikely to be embedded in routine practice in a non-hospital and non-research environment; this extends to whether equivalent PDC personnel can be found across facilities in a real-life larger study. Knowledge translation principles require affordability, feasibility and consideration of risks, not just evidence of technical effectiveness. It's acknowledged that this study aims to find out "how to optimise the implementation of complex interventions in nursing homes" but I would warn that the	The collection of individual level and system level data and the expert adjudication panels is not part of the intervention. This data is collected a part of the pilot study process to see which data and which outcomes would be best selected for a definite trial. This is clearly stated in the study aims on page 4.

excessive demands on staff carry the risk of turning the multicomponent intervention into an academic exercise with limited applicability in a chaotic and under-resourced real-life world. The discussion should include this as a limitation of the proposed study.